# Impact of Frailty on the Relationship between Blood Pressure and Cardiovascular Diseases and Mortality in Young-Old Adults

**DOI:** 10.3390/jpm12030418

**Published:** 2022-03-08

**Authors:** Sohyun Chun, Kyungdo Han, Seungwoo Lee, Mi-Hee Cho, Su-Min Jeong, Hee-Won Jung, Ki-Young Son, Dong-Wook Shin, Sang-Chol Lee

**Affiliations:** 1International Healthcare Center, Samsung Medical Center, Seoul 06351, Korea; sohyun.chun@gmail.com; 2Department of Family Medicine, Sungkyunkwan University School of Medicine, Seoul 06351, Korea; smjeong.fm@gmail.com; 3Department of Statistics and Actuarial Science, College of Natural Sciences, Soongsil University, Seoul 07040, Korea; 4Department of Biomedicine & Health Sciences, College of Medicine, The Catholic University of Korea, Seoul 06591, Korea; ghdk32@naver.com; 5Samsung C&T Medical Clinic, Kangbuk Samsung Hospital, Sungkyunkwan University School of Medicine, Seoul 03181, Korea; jom9990207@gmail.com; 6Division of Geriatrics, Department of Internal Medicine, Asan Medical Center, University of Ulsan College of Medicine, Seoul 05505, Korea; dr.ecsta@gmail.com; 7Department of Family Medicine, Asan Medical Center, University of Ulsan College of Medicine, Seoul 05505, Korea; mdsky75@gmail.com; 8Supportive Care Center, Samsung Comprehensive Cancer Center, Samsung Medical Center, Seoul 06351, Korea; 9Department of Digital Health, Samsung Advanced Institute of Health Science and Technology (SAIHST), Sungkyunkwan University, Seoul 06351, Korea; 10Department of Clinical Study Design and Evaluation, Samsung Advanced Institute of Health Science and Technology (SAIHST), Sungkyunkwan University, Seoul 06351, Korea; 11Division of Cardiology, Department of Internal Medicine, Samsung Medical Center, Sungkyunkwan University School of Medicine, Seoul 06351, Korea; sc.lea@samsung.com

**Keywords:** blood pressure, cardiovascular disease, heart diseases, aging, mortality, frailty

## Abstract

The optimal blood pressure (BP) target in older people according to frailty status remains uncertain. This article investigates how frailty affects the association between BP and cardiovascular diseases or mortality, specifically in young-old adults. A retrospective cohort was created for 708,964 older adults with a uniform age of 66 years. The association between BP and myocardial infarction (MI), stroke, or mortality was analyzed using Cox proportional hazards models. The Timed Up and Go test (TUG) was used as a measure of physical frailty. Mean follow-up was 6.8 years, detecting 38,963 (5.5%) events. There was a linear association between increasing systolic BP (SBP) or diastolic BP (DBP) and increased risk of incident MI and stroke, compared to the reference BP (SBP, 110–119 mmHg or DBP, 80–89 mmHg). The risk patterns with high BP remained similar in each TUG group (<10, 10–14, or ≥15 s). A similar pattern of increased risks was found in those who took antihypertensive drugs and who did not, however they were more pronounced in those who did not. The findings support the need to achieve the same BP target in young-old adults with or without frailty to lower the risk of MI, stroke, and mortality.

## 1. Introduction

Hypertension is the most common chronic disease in older adults and is a major risk factor for cardio-cerebrovascular disease, peripheral vascular diseases, cognitive disorders, and mortality. Despite having the highest prevalence and the greatest risk for cardiovascular (CV) morbidity and mortality, the prognosis for high blood pressure (BP) remains uncertain in the older population due to the heterogeneity of this group. Furthermore, management of high BP is hindered by other common geriatric problems, including frailty.

Frailty is an aging-related syndrome of physiological decline, multisystem dysfunction, and susceptibility to adverse consequences. Substantial evidence from observational studies has demonstrated that frailty can attenuate or even inverse the association between higher BP and mortality in older ages (Appendix A) [1,2,3,4,5,6,7,8]. In the Health and Retirement Study (mean age, 74.4 years), higher BP (≥150/90 mmHg) was associated with a higher mortality rate among participants with normal grip strength or walking speed but not those with frailty [8]. In a Mediterranean population study (mean age, 76.7 years), among slower walkers, normal or high BP (systolic BP [SBP], 120–139 mmHg or ≥140 mmHg) was associated with a decreased risk of mortality compared to lower BP [1]. In a Latino cohort study (mean age, 70.7 years), higher BP (≥160 mmHg SBP) was associated with elevated CV mortality risk among fast-walking participants but not those with slow or medium walking speed [3].

Despite the evidence from observational studies, randomized trials have demonstrated a beneficial effect of antihypertensive therapy and tight BP control in very old adults; for example, the Hypertension in the Very Elderly Trial (HYVET) showed the beneficial effect of antihypertensive treatment versus placebo on cardiac mortality and several other CV outcomes in adults in their 80s [9,10]. Furthermore, the Systolic Blood Pressure Intervention Trial (SPRINT) reported benefits with a lower BP goal of SBP < 120 mmHg in patients older than 75 years [11]. Most recently, a Chinese trial that involved 8500 hypertensive, older patients (age range, 60–80 years; mean age, 66 years) at high CV risk found that intensive treatment (SBP, 110–130 mmHg) resulted in a reduced incidence of CV events than conventional treatment (SBP, 130–150 mmHg) [12]. The post-hoc analyses of HYVET [13] and SPRINT [14] did not find evidence that frailty influenced treatment outcomes and concluded that antihypertensive treatment strategies and goals in frail older patients should be similar to those in the fittest subgroups of patients. Based on these studies, Hypertension Canada removed both advanced age (i.e., ≥80 years) and frailty as considerations for caution when deciding on intensive therapy from their 2017 guidelines for treatment of hypertension in adults.

Despite the accumulating studies, the optimal target BP in older individuals remains controversial. Furthermore, most studies assessing frailty are conducted using adults older than 70 years, and evidence is lacking for relatively younger older (young-old) adults (65–74 years old) who can be active and functionally capable and for whom functional decline might have a greater clinical impact. In this study, we aimed to investigate the relationships between varying BP and CV outcomes and mortality in relation to frailty in young-old adults 66 years of age.

## 2. Materials and Methods

### 2.1. Data Source and Study Subjects

Korea has a mandatory health insurance scheme where the Korean National Health Insurance Service (NHIS) provides healthcare coverage to all citizens and a mandatory biennial general health screening program to people aged 40 years and older. For people who are 66 years of age, assessment of geriatric physical and cognitive function is added to the basic health screening items as a program called the National Health Screening Program for Transitional Ages (NSPTA). The Korean NHIS maintains a database of demographic factors of age, sex, income, death date, utilization of medical facilities, and health screening results.

This study included a large-scale retrospective cohort including all individuals who were 66 years of age and participated in the NSPTA program from 2009 to 2012 (*n* = 909,489). Any subjects who were unable to undergo the geriatric physical function test were excluded (*n* = 17,772). Subjects who were diagnosed with MI, stroke, end-stage renal disease, and cancer before the health screening day (*n* = 179,787) and those with missing records on covariates were excluded (*n* = 129,705). The final population comprised 708,964 subjects.

This study was approved by the International Review Board of Samsung Medical Center (no. SMC202011077). Since the data were provided anonymously by the Korean National Health Insurance (KNHI), the study was exempt from the need to gather informed consent from the subjects.

### 2.2. BP Measurements

Blood pressure was measured using a standard protocol provided by the NSPTA manual. The measurement protocol was to measure brachial BP after five minutes of rest in a sitting position. The study population was divided into eight groups according to observed SBP (<100, 100–110, 110–119, 120–129, 130–139, 140–149, 150–159, and ≥160 mmHg) and seven groups according to observed DBP (<60, 60–69, 70–79, 80–89, 90–99, 100–110, and ≥110 mmHg).

### 2.3. Physical Function Test

The Timed Up and Go test (TUG) was used as a measure of physical performance to determine physical frailty. This test has been used as a basic functional assessment for vulnerability in older adults [15,16] and correlates well with other frailty measures in the Korean population [17]. Subjects were categorized into three groups according to TUG results: <10 s, 10–15 s, and ≥15 s.

### 2.4. Study Endpoints

The endpoints of the study were newly diagnosed MI, stroke, or total mortality. Myocardial infarction was defined as International Classification of Diseases, 10th revision (ICD-10) code I21 or I22 during hospitalization or at least two records of each code. Stroke was defined as ICD-10 code I63 or I64 during hospitalization with claims for brain magnetic resonance imaging or brain computerized tomography. Information on date of death was obtained from the National Statistical Office in Korea. Subjects were followed from NSPTA screening date (baseline) and censored on the day of occurrence of MI, stroke, or death or until the last follow-up date (31 December 2017), whichever occurred first.

### 2.5. Covariates

Body mass index (BMI) was calculated as body weight (kg) divided by height squared (m^2^). Information on smoking status, alcohol consumption, and regular exercise was obtained through questionnaires. Smoking status was classified as none, past, or current smoker, and alcohol consumption was classified as none, mild, or heavy (≥30 g of ethanol/day). Regular exercise was defined as performing strenuous physical activity for at least 20 min more than once a week. Income level was classified into medical aid and quartiles for the rest.

Hypertension was defined using the following criteria: (1) ICD-10 code for hypertension (I10–I11) with at least one prescription of an antihypertensive agent or (2) SBP/DBP of at least 140/90 mmHg at health screening. Diabetes was defined using the following criteria: (1) ICD-10 code for diabetes (E10–E14) with at least one claim of anti-diabetic medication or (2) fasting glucose level ≥ 126 mg/dL. Glomerular filtration rate (GFR) was estimated using the Modification of Diet in Renal Disease (MDRD) equation, and chronic pulmonary obstructive disease was noted based on diagnostic codes.

### 2.6. Statistical Analyses

Differences in characteristics between subjects with and without identified outcomes were evaluated using the t-test and Chi-square test.

Cox proportional hazards analyses were performed to estimate hazard ratios (HRs) and 95% confidence intervals (CIs) for MI, stroke, and total mortality according to BP levels and TUG scores. All models were adjusted for the following variables: sex, BMI, smoking, alcohol drinking, regular exercise, income, antihypertensive medication use, diabetes, chronic pulmonary obstructive disease, GFR, and hemoglobin level. The analysis was carried out in two steps. First, subjects with the reference BP and TUG < 10 s were set as the reference group and used to investigate the risk of the endpoints based on BP levels and physical frailty. Then, to study the independent effects of BP on the endpoints among subjects with similar frailty levels, another analysis was carried out using the reference BP group as the reference within each TUG group. The analyses were stratified by the use of antihypertensive drugs. All *p*-values were two-tailed, and a *p*-value < 0.05 was considered statistically significant. All statistical analyses were performed using the SAS version 9.4 software program (IBM Corporation, Armonk, NY, USA).

## 3. Results

### 3.1. Baseline Characteristics

During a mean follow-up period of 6.75 years, 5.5% of the study subjects (*n* = 38,963) experienced MI (*n* = 13,827), stroke (*n* = 26,936), or mortality from any cause (*n* = 33,996). Table 1 summarizes and compares the baseline risk factors between groups with and without the endpoint events. The event group had a higher proportion of male subjects than the group without events. At baseline, there were more current smokers and heavy drinkers (*p* < 0.001), more subjects with diagnosis of diabetes, and higher proportion of subjects taking antihypertensive drugs (*p* < 0.001) in the event group. Subjects in the event group also had slightly higher BMI, SBP (130.7 vs. 128.4 mmHg), and DBP (79.0 vs. 77.9 mmHg) levels and higher fasting blood sugar level compared to those without endpoint events. The low-density lipoprotein (LDL) level was not significantly different at baseline between the two groups.

### 3.2. Risk of MI, Stroke, and Mortality by BP in all Subjects

The incidence rates (per 1000 person-years) of MI, stroke, and mortality generally increased across levels of SBP and DBP (Figure 1 and Appendix A). Compared to the reference BP (SBP, 110–119 mmHg or DBP, 70–79 mmHg), risk of MI and stroke increased with elevated BP, and the result was statistically significant from SBP 130–139 mmHg and DBP 80–89 mmHg. The adjusted HRs were generally higher for stroke than for MI. The mortality risk increased with increasing SBP and DSP beginning at SBP 140–149 mmHg (adjusted HR, 1.05, 95% CI, 1.01–1.10) or DBP 90–99 mmHg (adjusted HR 1.02, 95% CI 1.07–1.19). An elevated mortality risk was observed with the lowest SBP (<100 mmHg), suggesting a J-shaped pattern.

### 3.3. Risk of MI, Stroke, and Mortality by BP and TUG Results

Figure 2 represents the association between BP and incident MI, stroke, and mortality, stratified by TUG and compared to the reference BP (SBP, 110–119 mmHg or DBP, 70–79 mmHg) and normal TUG (<10 s) as the common reference group. The risk for MI and stroke increased in a linear fashion across the increasing levels of BP in all TUG groups. A J-shaped pattern by BP level was observed with mortality and was more prominent in abnormal TUG groups (Appendix A).

### 3.4. Risk of MI, Stroke, and Mortality by BP within a Group Stratified by TUG Results

Figure 3 (Appendix A) shows the association between BP and incident MI, stroke, and mortality, stratified by TUG group and with 110–119 mmHg SBP or 70–79 mmHg DBP as the reference group. The patterns were similar to that shown in Figure 2, and the HR values in each BP category are similar after stratifying by TUG performance.

### 3.5. Risk of MI, Stroke, and Mortality by BP and TUG Stratified by Antihypertensive Treatment

General patterns were not different between those who took antihypertensive drugs and those who did not (Appendix A). Increased risk for MI and stroke with elevated BP tended to be more pronounced in those who did not take antihypertensive drugs. Among those who took antihypertensive drugs, the HR at SBP 140–149 mmHg was similar to or slightly lower than that at SBP 150–159 mmHg for all endpoints and across all TUG strata, while SBP in the range of 100–139 mmHg was not associated with increased risk. A DBP < 100 mmHg was associated with greater mortality but not a higher risk for MI or stroke in subjects with the slowest TUG result.

## 4. Discussion

In this population-based study consisting of young-old adults of the same age (66 years), we found that increasing BP was associated with increased risks of incident MI, stroke, and mortality, and this relationship was observed regardless of frailty status. The excessive risks seen with elevated BP were found both in subjects who were and were not being treated with antihypertensive drugs, but the risks increased at a lower threshold among those who were not being treated.

### 4.1. Frailty and Cardiovascular Diseases (CVDs)

Findings from our study support the current evidence that frailty is associated with higher risk of CVD and mortality [2,18,19]. Compared to the normal TUG group, those in the abnormal TUG groups with similar SBP or DBP ranges showed higher incidence of MI, stroke, and mortality. Poor performance on TUG is an independent risk factor for CVD or mortality [20]. This relationship was explained by the shared pathophysiological pathways between sarcopenia and CVDs, such as atherosclerosis and arterial stiffness. Chronic inflammation and oxidative stress lead to atherosclerosis, which can manifest clinically as poor performance on TUG, or frailty, and can result in increased occurrence of CV events.

### 4.2. Blood Pressure Goals

In our study, we did not find evidence to support higher BP goals for adults with frailty. Several epidemiologic studies have reported that frail, older people might not benefit from tight BP control or observed inverse association between BP and mortality [21]. However, our results showed that SBP ≥ 120 mmHg or DBP ≥ 80 mmHg is associated linearly with greater incidence of MI, stroke, and mortality. When subjects with similar TUG results were compared, excessive risk with elevated BP was observed similarly and consistently in the abnormal TUG groups as well. This suggests that high BP does increase the risk of MI, stroke, and mortality in even relatively young-old adults with frailty. It also supports benefit to frail older adults from tight BP control and advocates for the appropriateness of the 2017 Canadian guideline revision, which removed frailty as a consideration for caution when deciding on the intensity of antihypertensive therapy. Frail older people have a higher risk of MI or stroke and should be regarded as a high-risk group in which tight BP control is necessary.

### 4.3. Elevated BP and CVDs

Our study also showed that the increased risk triggered by elevated BP was more prominent for stroke than for MI. This is consistent with previous studies suggesting a higher association between BP and stroke than between BP and MI. Although MI and stroke are characterized by some common aspects and similar known risk factors, different pathogenetic mechanisms lead to MI and stroke. Several population-based studies have shown that hypertension increases the relative risk of stroke to a greater degree than it does to MI [22,23,24,25]. The difference in risk factors between coronary disease and stroke was suggested for the reason: serum cholesterol being a strong risk factor for coronary events, but not for stroke, and high blood pressure and high BMI being more important risk factors for stroke than for coronary events.

### 4.4. Treated and Untreated High BP and Outcomes

Stratification of subjects using and not using antihypertensive drugs further reinforces our conclusion. The HRs for association of BP with endpoints were more pronounced for subjects not on antihypertensive drugs than for those already on drugs. This might be partly due to the lower absolute risk in the reference group (i.e., risk in those not using antihypertensive drugs vs. risk in those using antihypertensive drugs). However, it also denotes the risk of untreated hypertension and signifies the importance of BP control.

Among subjects who were being treated with medication, while the association was not as pronounced as in those without medication, elevated BP was associated with MI and stroke risk, and the tight BP control group (SBP, 120–139 mmHg) did not demonstrate any higher risk (and probably showed a slightly lower risk) than the loose BP control group (SBP, 150–159 mmHg). This pattern was observed similarly in the abnormal TUG groups as well. While some guidelines acknowledged frailty as a consideration for caution when deciding on therapy for treatment of hypertension, our findings indicate that lenient BP targets should not be pursued in relatively young-old adults being treated for hypertension, even for those with frailty.

There is concern about an adverse effect of tight BP control in that it can lead to excess CV risk or mortality and be a possible source of the J-shaped association [26,27]. A recent post-hoc analysis of the SPRINT and Action to Control Cardiovascular Risk in Diabetes—Blood Pressure Intervention (ACCORD-BP) trial explored the lower limit of DBP among patients with controlled SBP < 130 mmHg and showed that lowering DBP < 60 mmHg was associated with increased risk of CV events compared to DBP 70–80 mmHg. In our study, the J-shaped association for BP was found only with mortality, particularly in the frail group. However, the elevated risk seen with low BP was attenuated when HR was calculated among frail subjects. Furthermore, this association was found in both treated and untreated subjects. This suggests that the increased mortality risk might not be an adverse effect of overtreatment but from the lower BP level per se. In addition, the risk of MI and stroke was not significantly elevated with the lowest BP, suggesting that low BP in old age might be due to other factors, such as orthostatic hypotension, malnutrition, dehydration, and other medications. The presence of low BP probably reflects these other medical conditions, leading to consequences during the life-course and, therefore, a higher mortality risk [28]. Clinicians should aim to avoid high BP rather than to avoid low BP by loosening treatment in older hypertensive adults with and without frailty.

### 4.5. Strengths and Limitations

While many older adults, especially young-old adults, function actively in the community and maintain physical fitness, they are often grouped simply as “older adults.” The uniqueness of this study is that it included adults 66 years of age only, allowing us to explore the association between BP and CVD and mortality in this distinctive population. The uniform age of the study subjects allowed us to control the age factor, which is one of the most important factors in research. The study is population-based and included more than 0.7 million older adults and might be a more representative sample than that typically recruited for clinical trials.

Despite its strengths, there also are several limitations to the analysis. First, only baseline BP levels were analyzed, and their changes over time were not considered. There are potential residual confounders, such as other drugs prescribed for the comorbidities of hypertension or variable compliance with antihypertensive drug usage. Moreover, although the TUG is a reliable, valid tool for assessing physical function, it cannot replace a full physical function test, and there could have been over- or underestimation of the number of frail subjects when we tried to assess frailty with TUG only. Lastly, although having a uniform age for study participants was a uniqueness of study, we were not able to explore young-old adults of other age ranges, and this can limit the generalizability of our findings.

## 5. Conclusions

In conclusion, there was a linear association between increasing SBP or DBP and increased risk of incident MI, stroke, and mortality in young-old adults, and the association was persistent regardless of frailty status. The elevated risk seen with higher BP was more pronounced in older adults who were frail than in those who were not. Data from this population-based study constitute robust evidence that supports the need for a common BP target in young-old adults with or without frailty to lower risk of MI, stroke, and mortality.

## Figures and Tables

**Figure 1 jpm-12-00418-f001:**
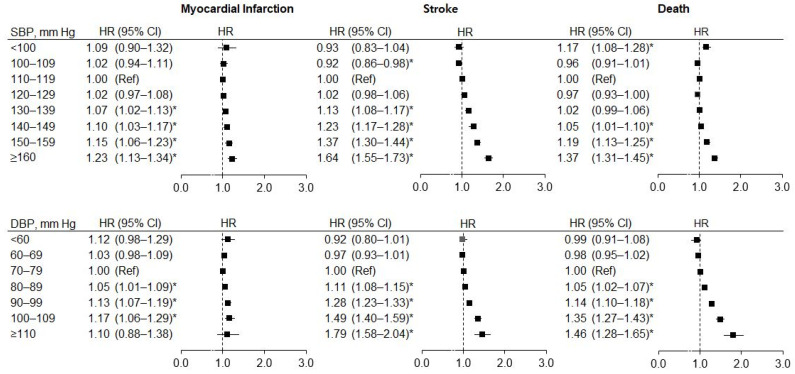
Hazard ratios for incidence of myocardial infarction, stroke, and death according to blood pressure in the total study population. Hazard ratios were adjusted for sex, BMI, smoking, hemoglobin level, alcohol drinking, regular exercise, income, antihypertensive drug use, diabetes, chronic kidney disease, and chronic pulmonary obstructive disease. Error bars indicate 95% confidence intervals. Asterisks indicate statistically significant values. SBP, systolic blood pressure; DBP, diastolic blood pressure; HR, hazard ratio; CI, confidence interval; Ref, reference.

**Figure 2 jpm-12-00418-f002:**
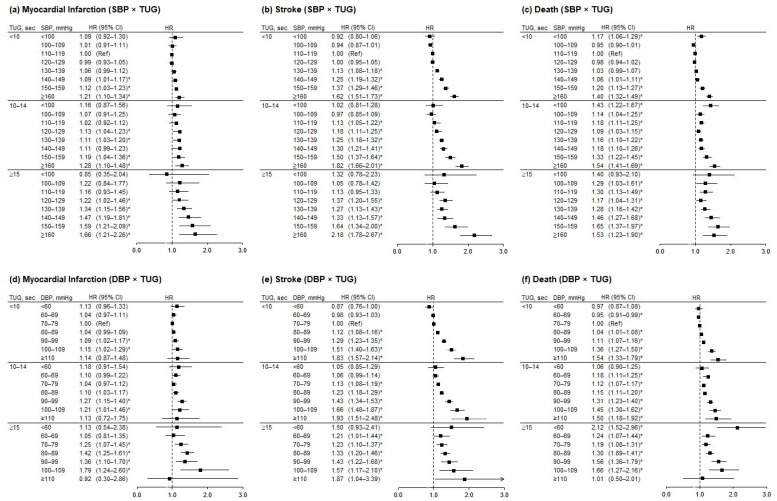
Hazard ratios for incidence of myocardial infarction, stroke, and death according to blood pressure, stratified by timed-up-and-go (TUG) test performance. Hazard ratios were adjusted for sex, BMI, smoking, hemoglobin level, alcohol drinking, regular exercise, income, antihypertensive drug use, diabetes, chronic kidney disease, and chronic pulmonary obstructive disease. Error bars indicate 95% confidence intervals. Asterisks indicate statistically significant values. TUG, timed-up-and-go test; SBP, systolic blood pressure; DBP, diastolic blood pressure; HR, hazard ratio; CI, confidence interval; Ref, reference.

**Figure 3 jpm-12-00418-f003:**
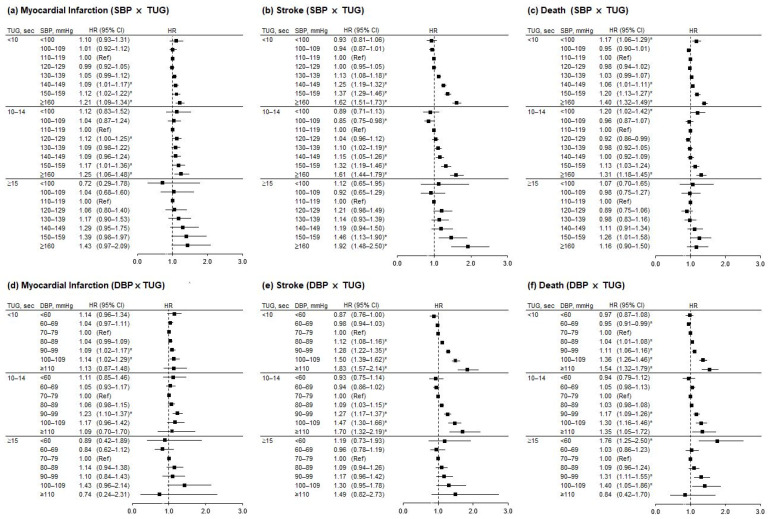
Hazard ratios for incidence of myocardial infarction, stroke, and death according to blood pressure, using 110–119 mmHg SBP or 70–79 mmHg DBP as a reference within each timed-up-and-go (TUG) group. Hazard ratios were adjusted for sex, BMI, smoking, hemoglobin level, alcohol drinking, regular exercise, income, antihypertensive drug use, diabetes, chronic kidney disease, and chronic pulmonary obstructive disease. Error bars indicate 95% confidence intervals. Asterisks indicate statistically significant values. TUG, timed-up-and-go test; SBP, systolic blood pressure; DBP, diastolic blood pressure; HR, hazard ratio; CI, confidence interval; Ref, reference.

**Table 1 jpm-12-00418-t001:** Baseline characteristics of subjects with and without incident endpoints.

	MI, Stroke, or Death		
Characteristics	No (*n* = 670,001)	Yes (n = 38,963)	*p*-Value
Sex			<0.0001
Male	302,784 (45.2)	20,914 (53.7)	
Female	367,217 (54.8)	18,049 (46.3)	
Income			<0.0001
Basic Livelihood Security recipients	21,452 (3.2)	2114 (5.4)	
Q1 (low)	166,615 (24.9)	10,120 (26.0)	
Q2 (low-middle)	129,312 (19.3)	7739 (19.86)	
Q3 (high-middle)	179,599 (26.8)	10,329 (26.5)	
Q4 (high)	173,023 (25.8)	8661 (22.2)	
Smoking			<0.0001
None	472,141 (70.5)	24,198 (62.1)	
Past	110,171 (16.4)	6198 (15.9)	
Current	87,689 (13.1)	8567 (22.0)	
Drink			<0.0001
None	475,137 (70.9)	26,682 (68.5)	
Mild	167,022 (24.9)	10,067 (25.8)	
Heavy	27,842 (4.2)	2214 (5.7)	
Regular exercise (yes)	313,320 (46.8)	16,289 (41.8)	<0.0001
Antihypertensive drugs	270,767 (40.4)	19,176 (49.2)	<0.0001
Diabetes mellitus	120,441 (18.0)	10,662 (27.4)	<0.0001
eGFR, mL/min/1.73 m^2^			
Mean, SD	83.7 ± 30.3	82.05 ± 32.66	<0.0001
<30	4949 (0.7)	440 (1.1)	
30–59	70,110 (10.5)	5189 (13.3)	
≥60	594,942 (88.8)	33,334 (85.6)	<0.0001
COPD	85,691 (12.8)	5,939 (15.2)	<0.0001
BMI (kg/m^2^)	24.3 ± 3.0	24.4 ± 3.1	<0.0001
Hemoglobin	13.6 ± 1.4	13.8 ± 1.5	<0.0001
Fasting blood glucose, mg/dL	102.8 ± 24.8	108.5 ± 33.8	<0.0001
Systolic blood pressure, mmHg	128.4 ± 15.4	130.7 ± 16.3	<0.0001
Diastolic blood pressure, mmHg	77.9 ± 9.7	79.0 ± 10.2	<0.0001
Total cholesterol, mg/dL	199.2 ± 37.9	200.2 ± 39.7	<0.0001
HDL-cholesterol, mg/dL	54.2 ± 17.0	52.7 ± 18.7	<0.0001
LDL-cholesterol, mg/dL	118.7 ± 40.8	118.7 ± 41.2	0.8

## Data Availability

The data that support the findings of this study are available from Korean National Health Insurance Services but restrictions apply to the availability of these data, which were used under license for the current study, and so are not publicly available. Data are however available from the authors upon reasonable request and with permission of Korean National Health Insurance Services at https://nhiss.nhis.or.kr (accessed on 7 March 2022).

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
