# Peer review of "Impact of Frailty on the Relationship between Blood Pressure and Cardiovascular Diseases and Mortality in Young-Old Adults"

_jpm, 2022, doi:10.3390/jpm12030418_

Round 1

Reviewer 1 Report

This study focus on the impact of frailty on cardiovascular and mortality events in a large population cohort for 3 years. The authors define human subjects for the study as young-adults of 66 years of age. The measurements at the inclusion time include: blood pressure and physical function test. The endpoints considered are events such as myocardial infarction, stroke and mortality.

Overall, the experimental design, organization of the manuscript is fair and may provide novel insights. However, I recommend to improve some aspects of the data based on my comments below:

Methods: Income level classification might be clarified: eg Q4 corresponds to higher income range.

Results.

Table 1. Is there any potential explanation about the lack of differences in LDL? Please discuss.

Figure 2 and Figure 3. The labelling formats of events (MI, S, D) could be kept for better reading.

Discussion:  Population data should be put in perspective of personalized medicine approach, eg What kind of interventions/recommendations can be provided to manage patients at risk?

Reviewer 2 Report

Paragraph 4.5 "Strenghts and limitations"

The present study has an impressive data amount and solid statistics. However, as the authors themselves recognised, BP was only collected at baseline and this greatly diminish the impact of a paper whose aim is to investigate the relationship between BP and CV events and mortality over a prolonged period of time. 

Moreover, the "younger old adults" category includes patients ranging from 64 to 74 years old whereas the study sample is only represented by 66 years old patients. This could be a possible bias and should be cited among the study's limitations, not strenghts.
